# Data-driven Taylor-Galerkin finite-element scheme for convection problems

**Luciano Drozda**
Algo-COOP team, CERFACS
Toulouse, France
drozda@cerfacs.fr

**Pavanakumar Mohanamuraly**
Algo-COOP team, CERFACS
Toulouse, France
mpkumar@cerfacs.fr

**Yuval Realpe**
ORT Braude College of Engineering
Karmiel, Israel
YuvalRal@braude.ac.il

**Corentin Lapeyre**
Algo-COOP team, CERFACS
Toulouse, France
lapeyre@cerfacs.fr

**Amir Adler**
ORT Braude College of Engineering
Karmiel, Israel
adleram@braude.ac.il

**Guillaume Daviller**
CFD team, CERFACS
Toulouse, France
daviller@cerfacs.fr

**Thierry Poinsot**
CERFACS / IMFT Institut de Mecanique des Fluides de Toulouse, CNRS
Toulouse, France
poinsot@cerfacs.fr

## Abstract

High-fidelity large-eddy simulations (LES) of high Reynolds number flows are essential to design low-carbon footprint energy conversion devices. The two-level Taylor-Galerkin (TTGC) finite-element method (FEM) has remained the workhorse of modern industrial-scale combustion LES. In this work, we propose an improved FEM termed ML-TTGC that introduces locally tunable parameters in the TTGC scheme, whose values are provided by a graph neural network (GNN). We show that ML-TTGC outperforms TTGC in solving the convection problem in both irregular and regular meshes over a wide-range of initial conditions. We train the GNN using parameter values that (i) minimize a weighted loss function of the dispersion and dissipation error and (ii) enforce them to be numerically stable. As a result no additional ad-hoc dissipation is necessary for numerical stability or to damp spurious waves amortizing the additional cost of running the GNN.

## 1 Introduction

Computational Fluid Dynamics (CFD) is a key tool to accelerate the design of engineering systems, notably in the aerospace industry where prototyping and trial-and-error can be excessively costly (Hirsch [2007]). Fluid flows are governed by the Navier-Stokes (NS) equations giving rise to chaotic turbulent flows that can be daunting to solve for directly. Depending on the flow regime and the time and cost constraints, simplified forms of the underlying partial differential equations (PDEs) are often considered. In combustion and aero-acoustics, despite the need of high-fidelity description of the unsteady flow phenomena, resolving all length scales of the turbulent flow is prohibitively expensive,

35th Conference on Neural Information Processing Systems (NeurIPS 2021), Sydney, Australia.

and a filtered version of the NS equations is solved instead, such as in LES (Boudier et al. [2007], Boudet et al. [2005]).

Applications of Machine Learning (ML) to the modelling of fluid flows take various approaches to informing the problem with its governing PDEs. Raissi et al. [2019] proposed to approximate the fluid flow function of space and time by directly minimizing the PDE on a sample of observation points, a popular approach known as Physics Informed Neural Networks (PINNs). This is a powerful method for problems where multiple constraints compete, when parts of the problem are not well known (*e.g.* boundary conditions in hydrology, initial conditions in weather predictions...), but for well-defined problems it is inefficient compared to direct resolution. Bar-Sinai et al. [2019] directly optimized the discretization technique for the PDE. Starting from Burgers' equation, a prototypical non-linear equation indicative of compressible flows, a strategy was derived to optimize the dissipative and dispersive behaviors of the numerical solution locally in the flow by means of accurate gradient estimations.

The present work describes a data-driven strategy to achieve a NS solver with locally-adaptable gradient estimation inspired by Bar-Sinai et al. [2019], but leveraging a high-accuracy, high robustness unstructured numerical scheme, namely the Two Level Taylor-Galerkin-Colin (TTGC-$\gamma$) scheme, originally proposed by Colin and Rudgyard [2000]. It is used in a myriad of simulations of reactive flows (Moureau et al. [2004], Schmitt et al. [2017], Vignat et al. [2021]) for its overall lower dissipation error and computational cost in comparison to previous schemes from the Taylor-Galerkin (TG) family like Euler-Taylor-Galerkin (by Donea [1984]) and Two-step Taylor-Galerkin (by Quartapelle and Selmin [1993]). The scheme we present is ML-TTGC: a variation of the TTGC-$\gamma$ scheme subject to ML-predicted local parameters. Optimal values of ML-TTGC parameters are found by a neural network model relying on Graph Nets (as proposed by Battaglia et al. [2018]). The main motivation for choosing a graph neural network architecture is its inherent capacity to represent any mesh and solution field as parts of a simple graph object, making the approach easily extendable to higher dimensions. We demonstrate the capabilities of ML-TTGC in addressing known issues of the original scheme by means of experiments conducted in a fully differentiable solver for 1D convection.

## 2   Locally tunable TTGC-$\gamma$ scheme

The problem of convecting a quantity $u := u(x, t)$ over time and space follows the PDE

$$u_t = -c u_x, \tag{1}$$

where $c$ is the convective speed and given initial condition $u(x, 0) = u_0(x)$. The sub-scripts $(\cdot)_t = \frac{\partial}{\partial t}$ and $(\cdot)_x = \frac{\partial}{\partial x}$ denote the partial derivatives with respect to the time and space, respectively. Colin and Rudgyard [2000] proposed the single parameter TTGC-$\gamma$ scheme:

$$\tilde{u}^n = u^n + \alpha \Delta t u_t^n + \beta \Delta t^2 u_{tt}^n, \tag{2}$$
$$u^{n+1} = u^n + \Delta t \tilde{u}_t^n + \gamma \Delta t^2 u_{tt}^n, \tag{3}$$

where $\alpha = \frac{1}{2} - \gamma$, $\beta = \frac{1}{6}$, and $\gamma$ is a free parameter that controls the dissipation at high wave-numbers. TTGC-$\gamma$ is third-order accurate on regular meshes. The numerical method is stable under a condition expressed in terms of the Courant-Friedrichs-Lewy (CFL) number, defined as $N_c := h^{-1} c \Delta t$, where $h$ is the cell size. It was shown that the stability of TTGC-$\gamma$ is ensured for a CFL number that lies in the interval $[0.4, 1.0]$ for $\gamma \in [0, 1.0]$. An important problem not addressed by the authors is the determination of the optimal $\gamma$ considering a range of $N_c$ values in a given mesh. They use a single global optimal value for $\gamma$ for a fixed $N_c$. In irregular meshes, even if the convection speed $c$ and time step $\Delta t$ are kept constant, the local CFL number of the finite-element will differ from one element to another due to variation in local cell size.

Note that non-linear convective equations (e.g., Burgers' equation) give rise to variable convective speeds across elements, even on regular meshes (equally spaced). Therefore, setting a *local* (i.e., per element) optimal value of $\gamma$ becomes critical to improving the performance of TTGC-$\gamma$. Using a global optimal $\gamma$ as in the original formulation demands additional artificial dissipation when solving non-linear systems with shock or jump discontinuities on irregular meshes (Roux et al. [2010]).

## 3 ML-TTGC numerical method

Numerical solution to PDEs are usually obtained on a discretized domain or mesh, which are represented as a union of nodes (where one defines the solution) connecting to form a closed volume of elements. As indicated by Pfaff et al. [2021], such a structure can be assimilated as an ensemble of vertices and edges in a graph, making Graph Neural Networks (GNNs) good candidates for handling problems in mesh-based simulations.

Fields to be estimated by GNN models are usually placed at graph's vertices. Since the local optimal $\gamma$, a quantity we wish to predict in our study is defined element-wise, a natural choice for the ensemble $V$ of vertices is the set of finite-elements in the physical mesh, i.e. $V := \cup_i e_i$. Each pair of elements $\{e_i, e_j\} \in V$ that uniquely shares a common face naturally becomes an edge connecting these two vertices of the GN. This is equivalent to using the *dual* mesh as our graph, as shown in Fig. 1. Finally, each graph vertex has $e_i := [u_i, u_{i+1}]^\top$ (1D) as attribute, while each graph edge holds $\epsilon_{ij} := [N_{c,i}, N_{c,j}]^\top$ (1D) as feature vector.

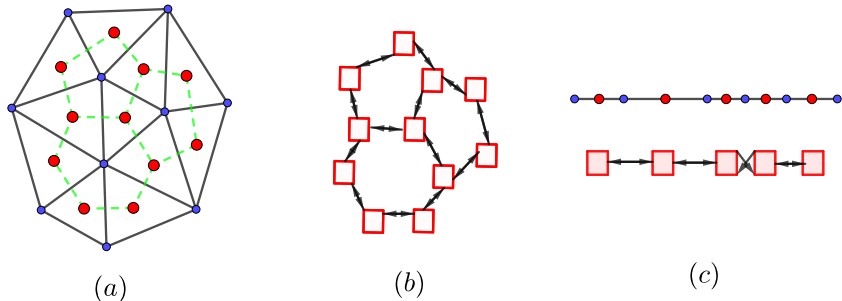

$(a)$ $(b)$ $(c)$

Figure 1: (a) Unstructured primal/dual mesh in 2D, (b) GN surrogate in 2D based on the dual mesh and (c) 1D primal/dual mesh (top) and GN based on the dual mesh (bottom); arrows denote the graph message-passing, red dots are dual graph vertices and blue dots are mesh nodes and green dotted lines are dual mesh edges.

We use an Encode-Process-Decode GN architecture illustrated in Fig. 2, which is inspired from the work of Pfaff et al. [2021]. The network is trained to map a solution state $u^n$ to reference per-cell values of $\gamma$. They are found via *a priori* optimization of locally tunable TTGC-$\gamma$ over one time step $(u^n \to u^{n+1})$ using a Sequential Least Squares Programming (SLSQP) gradient-based optimizer provided in the NLopt library. Since the optimized $\gamma$ values exhibit high temporal correlation, we included the $\gamma^{n-1}$ (value from previous time step) as an additional input attribute of the GN vertices. It is worth noting that the Decoder's output is re-scaled using a sigmoid function so that $\gamma^n$ does not violate the stability criteria for the locally tunable TTGC-$\gamma$, which is key to the generalizability of ML-TTGC.

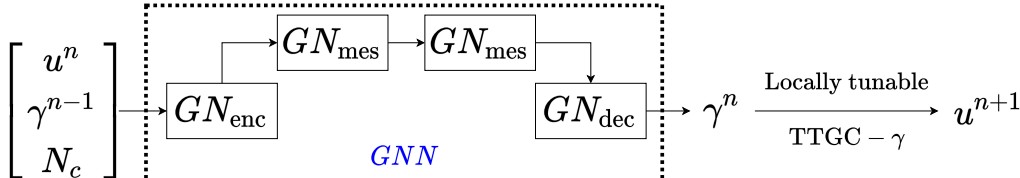

Figure 2: ML-TTGC architecture. The GNN instances trained for the studies later presented use an encoding, two message-passing and a decoding GN blocks, which update the vertices and edges attributes using MLPs with a hidden feature space of size 32. They have a total of $10,637$ trainable parameters each.

## 4 Results

We present two studies comparing performances of ML-TTGC with TTGC-$\gamma$ using a prescribed value of $\gamma = 0.01$. As stated by Colin and Rudgyard [2000], TTGC-$\gamma$ is unstable for negative values

of $\gamma$. Besides, they have found that values in the interval $[0, 0.05]$ give a much less dissipative scheme than other schemes from the TG family over the whole frequency spectrum for small CFL numbers (as per numerical results at CFL = 0.1) whereas it is comparable to them for high CFL numbers (provided tests at CFL = 0.7) and long to intermediate wavelengths.

In both studies we numerically solve the convection of a wave packet initial condition of the form:

$$u(x, 0) = \sin\left[k_0(x - x_0)\right] \exp\left[-\alpha(x - x_0)^2\right] \qquad (x \in [0, 1]) \tag{4}$$

where $k_0$ is the central frequency, $\alpha$ is the spectral spread and $x_0$ is the spatial location of the peak of the wave packet.

The first test case is the propagation of a low-frequency wave packet on highly irregular meshes. The GNN $\gamma$ predictor in ML-TTGC was trained on a set of $1024$ samples from $128$ optimized time-steps of waves at $(\mathrm{CFL}, k_0 h) = \{0.8, 0.9\} \times \{0.1, 0.4, 0.7, 1.\}$, where $h$ is the average mesh size of the irregular meshes used in the training (generated by uniform random perturbation proportional to $h$). Fig. 3(a) shows the numerical solution of the ML-TTGC and TTGC-$\gamma$ scheme for the convection problem (at time step 135) of a wave packet at $(\mathrm{CFL}, k_0 h) = (0.85, 0.5)$. Note that this wave packet is not part of the training data set and was deliberately chosen outside the set to demonstrate robustness. We see that the amplitude of the spurious oscillations (*wiggles*) is significantly reduced in ML-TTGC compared to the TTGC-$\gamma$ scheme where amplitudes of the spurious waves are quite significant. In reactive flow simulations, species transport is modelled using a convection-dominated equation and such numerical wiggles can lead to unphysical negative mass fractions. Therefore it is critical to mitigate them in the numerical solution. This is most commonly achieved by adding artificial dissipation, which significantly increases the solver runtime.

Our second test case concerns the propagation of high-frequency wave packets on regular meshes. We separately trained ML-TTGC on a set of $1024$ samples from the solution sets of wave-packet convection at $(\mathrm{CFL}, k_0 h) = \{0.5, 0.6\} \times \{1.1, 1.4, 1.7, 2.\}$. Fig. 3(b) illustrates the high dissipation of TTGC-$\gamma$ (typical at such range of wavenumbers), which contrasts with ML-TTGC's ability to preserve the wave's amplitude for longer time scales.

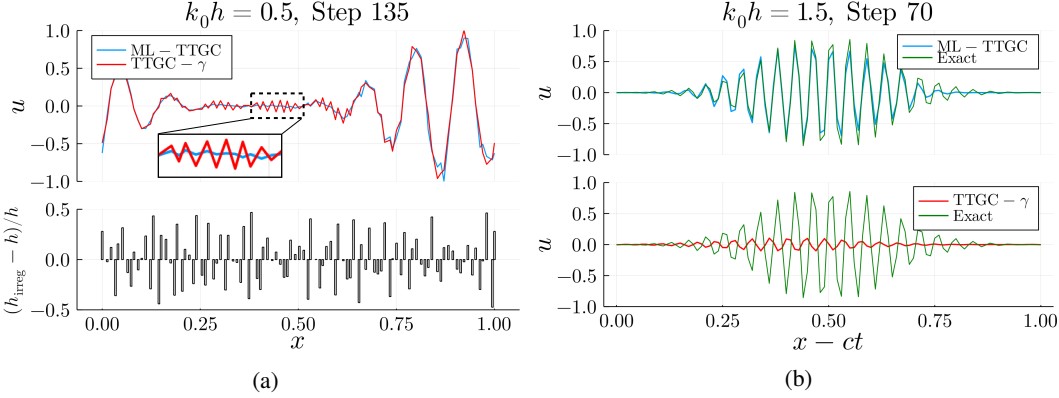

Figure 3: (a) Top: low-frequency wave packet convection over an unequally spaced mesh. Bottom: relative deviation of cell sizes from the average value. (b) Time step n. 70 of wave at $(\mathrm{CFL}, k_0 h) = (0.55, 1.5)$ (data is not part of the ML-TTGC training set). ML-TTGC and analytical solution (top); TTGC-$\gamma$ and exact solution (bottom).

A complementary approach for studying the GNN capabilities consisted on training on the exact solutions as opposed to using pre-optimized $\gamma$ as ground-truth. During training, the GNN along with the solver predict series of 10 time steps from initial conditions at different wavenumbers $k_0 h$ and a constant CFL value of 0.5. Fig.4(a) depicts two of such trained models showing their generalization capabilities (regarding predictions on a $k_0 h$ spectrum, on regular meshes). In general, models trained on relatively low wavenumbers ($k_0 h < 0.75$) generalize better: an incident wave out of the $k_0 h \in [0.10, 0.90]$ interval results in reliable predictions (steady MSE over sufficient time steps). Fig.4(b) is an example of further expanding the range of reliability by adding a high wavenumber sample into the training set, hinting at an optimal mix of wavenumbers for which the range of reliable predictions is maximized.

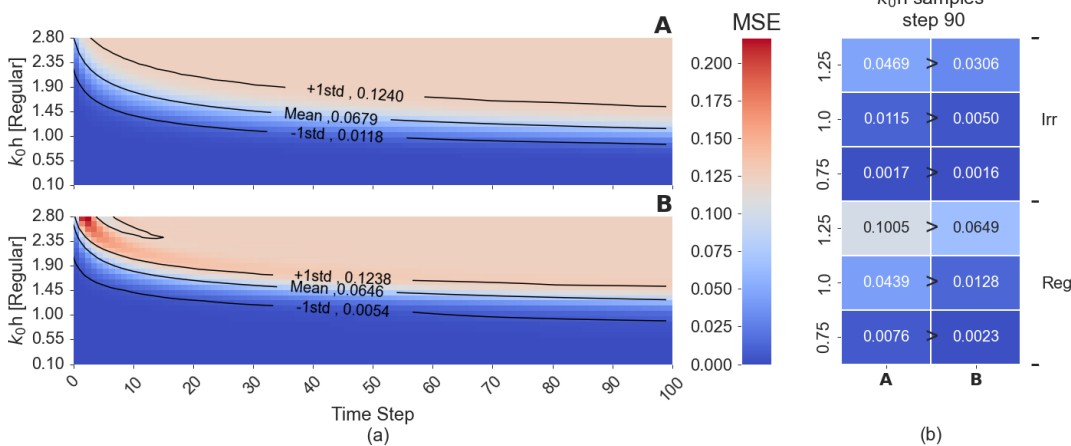

Figure 4: Comparison of the generalization capabilities of models A and B, evaluated for a set of incident $u(x, t; k_0)$ varying in $k_0$ and mesh regularity. Models A and B were trained on $(k_0 h) \in \{0.25\}$ and $(k_0 h) \in \{0.25, 1.0\}$, respectively. (a) Time evolution of the predicted solution's MSE for a dense sample of $(k_0 h) \in [0.1, 2.8]$. (b) specific showcase of incident $u(x, t; k_0)$ for $(k_0 h) \in (0.75, 1.0, 1.25)$ varying in regularity (Irr/Reg stands for irregular/regular mesh).

## 5 Conclusions

We introduce ML-TTGC, a data-driven finite-element method that addresses some critical limitations of the TTGC-$\gamma$ scheme used in LES. ML-TTGC successfully damps spurious oscillations and preserves amplitudes, even of high-frequency waves, for the linear convection problem. In future work, we plan to expand the GNN-based architecture and extend our scheme to 2D/3D edge-based finite-element methods for solving non-linear problems.

## Acknowledgments and Disclosure of Funding

This research was partly supported by the Ministry of Science & Technology, Israel.

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
