# OpenReview forum: "Data-driven Taylor-Galerkin finite-element scheme for convection problems"
_NeurIPS.cc/2021/Workshop/DLDE — DLDE Workshop -- NeurIPS 2021 Poster_

### Official Review · Reviewer_kTJf · 2021-09-30
**Nice contribution with good technical merits**

**Confidence:** 3

**Review:**

The authors propose a method for large-eddy simulations that improves on previous methods by damping spurious oscillations and preserving amplitudes, even of high-frequency waves, for the linear convection problem.

Pros:

- Very good presentation and well written paper
- Interesting problem with good real world impact, though I confess, not one I am very familiar with
- Nice results

Cons:

- Would be nice to know if this works in higher dimensions
- L80, would be a bit friendlier to define SLSQP before using the acronym

**Score:**

3: Good paper

---

### Official Review · Reviewer_DUbC · 2021-10-01
**A well written paper improving on the Taylor-Galerkin method for large-eddy simulations**

**Confidence:** 3

**Review:**

The authors introduce a new method to run large-eddy simulations. They adapt the Taylor-Galerkin finite-element method, which requires a global hyperparameter $\gamma$. Their contribution is to locally tune $\gamma$ using a graph neural network. They test their method on 1D simulations and their method improves on the standard Taylor-Galerkin method.

I have a few minor points to clarify:

- I may be wrong on this, but I believe there is a small error in equations 2 and 3 describing the TTGC-$\gamma$ scheme. Equation 2 gives an equation for $\delta \tilde{u}^{n}$, however in the TTGC-$\gamma$ scheme, I think that $\tilde{u}^{n}$ is used as a temporary value in the scheme rather than something that changes. I believe equations 2 and 3 should read as (or be equivalent to) equations 19 and 20 in Colin and Rudygard 2020, https://linkinghub.elsevier.com/retrieve/pii/S0021999100965380.

- Is there a reason that a value of $\gamma=0.01$ is used for the baseline method in the experiments? Would other values produce significantly different results?


It would be great to see this method applied to higher dimensional systems in future work.





**Score:**

4: Very good paper

---

### Official Review · Reviewer_RcMd · 2021-10-06
**Interesting paper, more explanation needed**

**Confidence:** 4

**Review:**

The authors adapt the Taylor-Galerkin finite-element method, which requires a global hyperparameter $\gamma$, to a local one using a graph neural network. They test their method on 1D simulations and their method improves on the standard Taylor-Galerkin method.

Overall, it is a decent work but it is not easy to read as explanation is missing:

(1)
The $\delta u^n$ notation is introduced in equations 2 and 3 but never explained or used afterwards. The reader needs to read the references provided to understand these equations and how they are used in the current setup.

(2)
 The acronym CFL is never explained making it hard for a non-expert in computational fluid dynamics to follow.

The paper would benefit by having a section on the training of the graph neural network: the design of the cost function, the computational cost of the training, how the overall computational cost of the proposed method compares to that of simply using the current state of the art (a global $\gamma$).

Also, how does the number of parameters in the network scale with grid size (number of cells) and dimensionality of the problem. Are there scenarios where the number of parameters becomes huge and non-feasible for practical purposes?

In addition, the authors need to comment on real world applications where no other solvers can be used to provide data for the proposed method. How would they proceed to gather data for the current method if there is no ground truth or baseline readily available?

I believe there is a potential for great improvement and I would be happy to change my review to that of a very good, if not excellent, paper if the concerns outlined here are addressed.

**Score:**

2: Borderline paper

---

### Decision · Program_Chairs · 2021-10-15

**Decision:**

Accept (Poster)

**Comment:**

Reviewers generally considered this a very strong submission. One reviewer raised a few important concerns about notation, methodology, and impact. In particular, all reviewers were curious about the extension of this work to higher dimensions. Authors may wish to address these points before the workshop to maximize the impact of their poster.